# Health Promotion and Support Grounded in Interconnected Influences on Alcohol Use in Pregnancy

**DOI:** 10.3390/ijerph22081309

**Published:** 2025-08-21

**Authors:** Nancy Poole, Lindsay Wolfson, Ella Huber

**Affiliations:** 1Centre of Excellence for Women’s Health, E209-4500 Oak Street Box 48, Vancouver, BC V6H 3N1, Canada; lindsay.wolfson@gmail.com (L.W.); ellahuber95@gmail.com (E.H.); 2Canada FASD Research Network, P.O. Box 11364 Wessex PO, Vancouver, BC V54 0A4, Canada

**Keywords:** Fetal Alcohol Spectrum Disorders, pregnancy, women’s health, alcohol drinking, maternal health, social determinants of health, social stigma

## Abstract

There are a range of factors that influence alcohol use in pregnancy and create risk of fetal harm. However, limited research has articulated the multilevel nature of these influences and their entanglement. The purpose of this narrative review is to analyze the types of factors that influence alcohol use and consider what factors need to be addressed in future health promotion and intervention efforts. Six databases were searched using EBSCOhost articles published between January and December 2023 on alcohol use in pregnancy and Fetal Alcohol Spectrum Disorder (FASD) prevention. English-language articles were screened for relevance and a subset of articles exploring the prevalence, influences, and risk-factors associated with pregnancy were included for analysis. Thirty-two (*n* = 32) articles were included in the review and categorized into five key areas of influence on maternal alcohol use: (1) informational factors; (2) stress-related factors; (3) social determinant of health-related (SDoH) influences; (4) preconception- and prenatal-health-related factors; and (5) structural factors. Future efforts to reduce alcohol use in pregnancy should address these five categories of factors through non-judgmental, health-promoting, trauma-informed, harm-reduction-oriented, and culturally safe education, programming, and policy.

## 1. Introduction

Alcohol use during pregnancy poses significant risks for fetal, child, and maternal health. Risks to women’s health and healthy pregnancies resulting from alcohol use in pregnancy include increased risk of miscarriage, still birth, hemorrhage in late pregnancy, and preterm pre-labor membrane rupture [1,2,3]. Prenatal exposure to alcohol increases the risk of a fetus being small for gestational age, preterm birth, and low birth weight, and can lead to neurocognitive, behavioral, congenital, and emotional challenges and Fetal Alcohol Spectrum Disorder (FASD) [3,4]. FASD is a lifelong disability that affects the brain and body, resulting in the need for special supports in many different domains of daily life [5].

Alcohol use in pregnancy and FASD are common, however, understanding the true prevalence is hindered by challenges posed from stigma and barriers to accessing prenatal support and FASD diagnosis [6]. An international review of research published between 2000 and 2022 reported prenatal alcohol use prevalence to be 8.7% in the USA and up to 20% in European countries such as France and Germany, with a mean worldwide prevalence of 9.8% [7].

Women who use alcohol and other substances during pregnancy are highly marginalized, stigmatized, and impacted by a range of social and structural factors [8,9,10,11,12]. This creates barriers to access and engagement in needed health and social support [13,14,15], which is exacerbated by social and structural factors, such as experiences of racism, intimate partner violence, and lack of housing. Additionally, recent years have seen increasingly punitive policies in response to alcohol and substance use during pregnancy, criminalizing women and designating prenatal substance use as child abuse or neglect [16]. For these reasons, women who use alcohol and other substances during pregnancy remain an under-served and under-supported population.

Despite there being extensive research on alcohol use in pregnancy, its focus has often been on the individual considerations and responsibilities of mothers, rather than the responsibility of systems of care to provide support to mothers that takes into account the social and structural determinants of health that burden them [12,17,18]. A new paradigm is needed that explores the range and types of key factors that influence alcohol use in pregnancy and acts to address them in concerted ways. This narrative review investigates and categorizes the factors that influence alcohol use in pregnancy on multiple levels and considers what factors need to be addressed to improve practice and policy aimed at reduction and prevention. Understanding the range of factors that influence alcohol use in pregnancy is key to developing effective services that support maternal and fetal health and prevent FASD.

## 2. Methods

This narrative review is based on a subset of data collected as part of a 2023 academic literature search on alcohol use in pregnancy and FASD prevention conducted by researchers associated with the Prevention Network Action Team of the Canada FASD Research Network. This academic literature review is conducted every year to update those working on FASD prevention in Canada on the peer-reviewed English-language research on perinatal alcohol use and FASD prevention. The findings are mapped using the Four-Part Model [19] to capture the wide range of work that comprises FASD prevention. Six databases were searched using EBSCOhost (accessed 8 January 2024): Bibliography of Native North Americans, CINAHL (Cumulative Index of Nursing and Allied Health Literature) Complete, MEDLINE with Full Text, PsycINFO, Social Work Abstracts, and Urban Studies Abstracts.

### Data Analysis and Synthesis

Peer-reviewed English-language articles published in academic journals in 2023 were screened for relevance to alcohol use in pregnancy and FASD prevention. Articles were included in the larger academic literature search if they described the prevalence and factors associated with alcohol use in pregnancy, preconception, or FASD prevention interventions, or described other systemic or ethical considerations related to women’s alcohol use during pregnancy or FASD prevention efforts. Articles were excluded if they were not relevant (e.g., if they focused on FASD diagnosis or intervention or perinatal health, generally). A total of *n* = 104 articles were included that addressed the prevalence and influences on alcohol use during pregnancy, interventions at each of the four levels, and systemic, destigmatizing, and ethical considerations. Wolfson et al. [12] outlines the methods used for previous annual literature searches (2015–2021).

This review examines a subset of articles (*n* = 32) that explored the prevalence of, and influences and factors associated with, alcohol use during pregnancy. A narrative review method was used because of its subjectivist approach, which both allows for a summary of the research as well as interpretation and critique [20]. Using procedures for thematic analysis [21], including recommendations for narratively synthesizing qualitative and quantitative evidence [22], the authors identified all of the factors associated with alcohol use in pregnancy and categorized them into overarching categories of factors and influences associated with alcohol use during pregnancy. The five categories of influence identified in the thematic analysis emerged from discussions of this body of literature by the authors and researchers, service providers, policy analysts, and women with lived experience who have participated in the Prevention Network Action Team of the Canada FASD Research Network and aligned with the broader literature on the influences and factors associated with alcohol use in pregnancy. Focusing on a subset of the literature capturing research published in 2023 allowed the authors to identify the five themes and the articulation of all five categories of influence.

## 3. Results

Thirty-two (*n* = 32) articles were identified between January and December 2023 that explored the prevalence of, and influences and factors associated with, alcohol use during pregnancy. Five articles were excluded because they reported on the prevalence of alcohol use during pregnancy without identifying influences or factors. Of the twenty-seven (*n* = 27) included studies, most were cross-sectional (*n* = 15), followed by qualitative (*n* = 3), mixed-methods (*n* = 2), systematic reviews and meta-analyses (*n* = 2), other reviews (*n =* 1), case–control (*n* = 1), cohort (*n* = 1), meta-ethnography (*n* = 1), and a randomized controlled trial (*n* = 1).

Twenty-six factors were identified in the literature that were associated with alcohol use in pregnancy. These factors were categorized into five key areas of influence on maternal alcohol use: (1) informational factors; (2) stress-related factors; (3) social determinant of health (SDoH)-related influences; (4) preconception- and prenatal-health-related factors; and (5) structural factors. As identified in Table 1, few studies reported on only one key area of influence. The subsequent section will further describe each key area of influence, as well as how it was reported upon in the literature.

### 3.1. Informational Factors

Informational factors include women and people of childbearing age having clear, accurate, consistent, accessible, and non-judgmental/non-stigmatizing information about the risks of alcohol use in pregnancy [18,48]. Information may be provided through personal interactions (e.g., brief interventions; Level 2 FASD prevention), prevention campaigns, alcohol warnings and beverage labeling, and community educational and health-promotion strategies (e.g., Level 1 FASD prevention) [19]. Several informational factors were described that influence alcohol use in pregnancy, including individual and community-based knowledge of the risks of alcohol use in pregnancy [27,31] and the (in)ability to safely discuss and learn about the risks of perinatal alcohol use with health and social care providers [33].

Research from Australia [27], the USA [31], and South Africa [36] found that there is mixed messaging and inconsistent perceptions about alcohol use in pregnancy. Researchers from Western Australian conducted an online questionnaire with *n* = 435 women. In an open-ended question on the advantages and disadvantages of alcohol use in pregnancy, most (99.5%) women mentioned at least one disadvantage of alcohol use during pregnancy; views on the perceived risk of different levels of alcohol use and perceptions of the ‘typical’ person who drinks while pregnant varied. Forty-four percent (44%) of respondents suggested that there could be benefits of drinking during pregnancy, such as for social connection, relaxation, stress relief, or to maintain/regain sense of self. Perceptions around alcohol use in pregnancy were found to be influenced by normative beliefs that approved/disapproved of alcohol use during pregnancy, perceived behavioral control, and perceptions towards women who do/would use alcohol during pregnancy [27].

In a qualitative study examining the reasons for and obstacles to changing risky alcohol use among Latina women at risk of an alcohol-exposed pregnancy, it was also found that beliefs persisted that alcohol use during pregnancy is not risky. These beliefs, as well as social pressure, the social acceptability of alcohol, and the use of alcohol to manage mood, were obstacles to reducing alcohol use. However, women cited several reasons for changing risky alcohol use, including health-related issues, enacting positive parenting role models, reducing interpersonal conflict, having control over drinking, and avoiding harms such as legal problems [31].

A South African study seeking to understand pregnant women’s and healthcare providers’ perceptions of the acceptability, feasibility, and appeal of a community-based counselling program for pregnant women with alcohol problems found that financial stress, intimate partner violence, ambivalence about pregnancy, and lack of knowledge about the risks of alcohol use during pregnancy all contributed to alcohol use in pregnancy [36]. These findings illustrate how informational influences are linked to other categories and levels of influence. The authors noted the lack of recognition of alcohol as a harmful substance at the community level [36], highlighting the role of broader normative values on the ability to address or reduce perinatal substance use. The study identified the need for community-based service providers to build rapport and create an environment in which women feel comfortable disclosing alcohol use. This requires professional guidelines, referral pathways, and ensuring that messaging is supportive and non-judgmental.

The importance of providing information about risks while making it safe to share information about use was further noted in an Australian study of pregnant women at a prison in New South Wales [33]. At reception to the facility, the authors found that only 6% of women reported alcohol use; however, 23% of women reported using alcohol at the two-week follow-up. The authors suggested that women may have found disclosure in a less chaotic environment, with a specialist, and at a later time, to be easier than during the initial screening process at reception. The safety to discuss substance use may have been influenced by factors such as a lack of privacy or concerns about child custody after release, if their use was noted on their file. However, these delays in identifying alcohol use as well as the lack of awareness of pregnancy status can, in turn, contribute to a delay in the delivery of tailored substance and reproductive healthcare assessments and supports [33].

The informational factors influencing alcohol use in pregnancy highlighted above demonstrate the ongoing importance of education about the risks of alcohol use in pregnancy and the need for health promotion and prevention efforts to provide support, discuss reasons for change, counter negative stereotypes, and promote understanding toward women who do not, or are not able to, stop drinking in the perinatal period.

### 3.2. Stress-Related Factors

The use of alcohol to cope with stressors and mental health concerns is often identified as a key contributor to the continuing use of alcohol in pregnancy. Stress-related factors refer to early-life, one-time life event, and cumulative trauma and stressors that influence alcohol use [49]. These may include the use of alcohol to cope with the challenges of day-to-day life, adverse childhood experiences, intimate partner coercive control and violence, pressures and challenges related to parenting and child custody, pressure to drink from partners, friends, and greater society, and mental health challenges [18,50]. The stress-related factors described in articles published in 2023 from North America, Europe, and Africa included depression, anxiety, or serious psychological distress [23,26,35,43]; interpersonal/intimate partner violence [35,43]; social pressure to use alcohol [24]; parenting challenges [31]; interpersonal conflict [31]; and childhood trauma [35].

Several studies reported on the impact of mental health and other substance use on perinatal alcohol use. A nationwide Danish cohort study found mothers of children with heavily alcohol exposed births were more likely to have psychiatric diagnoses, substance use, tobacco use, and low educational level in comparison to those without heavy alcohol exposure [1]. A study from the USA examining the relationship between Serious Psychological Distress (SPD) and substance use among pregnant women found approximately 6% of the pregnant women who experienced SPD had higher rates of cigarette use, binge drinking, and cannabis use compared to pregnant woman who did not report experiencing SPD [26]. In another study from the USA, depressive and anxiety disorders, intimate partner violence, and intergenerational substance use were greater among women who used multiple substances (e.g., alcohol, cannabis, nicotine, and stimulants) compared to those who used predominantly alcohol or cannabis [43].

The confluence of stress-related factors with other key areas of influence on perinatal alcohol use was identified. For example, an Ethiopian study found that a family history of mental illness, depression, and anxiety, as well as SDoH-related factors (e.g., single marital status) and preconception- and prenatal-health-related factors (e.g., history of abortion and pre-pregnancy alcohol use) were statistically significant and associated with perinatal alcohol use [23]. Another Ethiopian study examining the risk factors associated with khat and alcohol use among pregnant women found that low educational level, pre-pregnancy alcohol use, unplanned pregnancy, history of abortion, poor social support, and mental distress were all associated with alcohol use during pregnancy [47].

The use of alcohol to manage or cope with anxiety, stress, or day-to-day life experiences was reported in several studies. In a South African study examining the distal risk factors for FASD, mothers of children with FASD were reported to be less happy, have suffered more childhood trauma and interpersonal violence, were more likely to drink alone or with a partner, and drank to deal with anxiety and tension [35]. In a USA study examining the reasons for and obstacles to changing risky alcohol use among Latina adults at risk of an alcohol-exposed pregnancy (*n* = 59), 36% of participants noted using alcohol as a means for relaxation and coping with stressors related to work and finances. Women also reported difficulty with managing their emotions and how that influenced both their alcohol use and their ambivalence to change [31]. These findings emphasize the centrality of understanding and addressing a continuum of mental health and wellness issues, alongside perinatal alcohol and other substance use [1,26].

### 3.3. Social Determinant of Health (SDoH)-Related Influences

SDoH refer to the social and economic conditions that impact our health and the health and social services that people have access to [51]. SDoH include income and social status, employment and working conditions, education, food security, physical environments, gender, culture, and race/racism, among others [51,52], many of which influence alcohol use in pregnancy [53]. SDoH-related factors that were associated with alcohol use in pregnancy included marital status [23,29]; socioeconomic [25,34,37,39], education [25,29]; and employment status [25,32]; age [25,37]; and food security [40].

Socioeconomic status was most cited as influencing perinatal alcohol use. However, socioeconomic factors were often combined with other categories of influences to contribute to alcohol use during pregnancy [31,33,36,40,47]. For example, in interviews with *n* = 28 women and community health workers in South Africa, financial stress, in addition to intimate partner violence, ambivalence about pregnancy, and lack of knowledge about the risks of alcohol use during pregnancy, were identified as important factors influencing perinatal alcohol use [36]. Similarly, in a study of *n* = 202 women from Uganda and South Africa, cohabitating with a partner, food insecurity, and perceived stigma about HIV status influenced both continued and new alcohol use [40].

A Canadian study that pooled data from multiple cohort studies to identify sociodemographic characteristics associated with alcohol consumption during pregnancy found that a higher income level was associated with any drinking during pregnancy [39]. The study also found that women’s alcohol use during pregnancy (both any use and binge drinking) was associated with drinking prior to pregnancy, smoking during pregnancy, and white ethnicity. A systematic review led by German researchers on the impacts of prenatal alcohol and nicotine on early childhood development found that an increased likelihood of alcohol consumption in pregnancy was related to better social support and older age [37].

Another German study found that below-standard antenatal care was more likely to be experienced among women with unplanned pregnancies and who had less education and lower income. The risk for below-standard care was also higher for those who smoked during pregnancy and used alcohol. In this study, higher income was negatively correlated with smoking during pregnancy but positively associated with alcohol use during pregnancy [34]. 

A retrospective study with *n* = 23,894 postpartum women found that a higher prevalence of use was found among women with greater social vulnerability, including those with less education, those belonging to lower economic classes, those with unintended pregnancies, those with a higher number of previous births, those with late entry to prenatal care, and those with an inadequate number of prenatal care consultations. Higher prevalence was also found among women who were not white, did not have a partner during pregnancy, did not have paid employment, and were treated in public services [25].

While multiple SDoH-related factors were reported, all studies had some connection to structural-level socioeconomic factors. For example, two South African studies reported that the lack of formal employment and opportunities for recreation impacted the prevalence of alcohol use during pregnancy. Had there been formal employment opportunities, women’s hopelessness and alcohol consumption could be better addressed [24,32]. The authors of these studies noted that the intersecting risks justify multiple levels of intervention (individual, community-level, and structural) in programming designed to support reducing/stopping substance use during pregnancy [40].

### 3.4. Preconception- and Prenatal-Health-Related Factors

Preconception- and prenatal-health-factors refer to the health risks that occur when planning a pregnancy and in the preconception period—which may or may not continue in pregnancy. These factors include, but are not limited to, preconception alcohol, tobacco, and other substance use, perinatal substance use, vitamin intake and nutritional deficiencies, and chronic health issues [54]. In 2023, the preconception- and prenatal-health-related factors identified as related to perinatal alcohol use were higher parity [25,35], preconception substance use [39,45], smoking during pregnancy [25], lower prenatal vitamin intake [30], and overall health status [31]. A number of studies also described preparedness to be pregnant and intention to use alcohol in pregnancy as relevant for those who are developing interventions to prevent prenatal alcohol exposure [28,37,39,47].

A study conducted by Australian researchers explored intentions to consume alcohol during pregnancy [28]. Women (*n* = 746) aged 20–45 in Australia and the UK who were currently pregnant, had previously been pregnant, or had future pregnancy intentions, were recruited through local Facebook pages and online parenting forums. The authors investigated whether priming participants with exposure to prototypes describing different alcohol use behaviors would influence future alcohol use intentions. Using the theory of planned behavior, a method to predict individuals’ alcohol use, the authors found that 30.72% of participants reported that they intended to drink a ‘small’ amount of alcohol while trying to get pregnant, and 7.63% reported intending to drink while pregnant. However, the most significant predictor of intention to use alcohol was positive attitudes towards alcohol use in pregnancy. The study findings emphasized that future prevention interventions need to attend to changing attitudes towards low-to-moderate alcohol use during pregnancy.

Prenatal alcohol use was also influenced by preconception alcohol and other substance use and prenatal tobacco use [25]. Several studies found that drinking pre-pregnancy impacted alcohol use during pregnancy [39,45,46]. For example, an Australian study found that frequent preconception tobacco use, cannabis use, and binge drinking in adolescence and young adulthood were strong predictors of continued use during pregnancy, both before and after pregnancy recognition and at one year postpartum [45]. Preconception alcohol use and reproductive health factors co-occur with all other categories of influences. For example, pre-pregnancy alcohol use, unplanned pregnancy, history of abortion, poor social support, low educational level, and mental distress were factors associated with alcohol and khat consumption during pregnancy [47].

Other preconception- and prenatal-health-related factors, such as prenatal vitamin intake, were also found to influence alcohol use in pregnancy. A study from South Africa examined the dietary intake of pregnant women in communities with high rates of FASD. The study found that most women were below the recommended amount for vitamins A, C, D, and E, choline, calcium, magnesium, zinc, and potassium in pregnancy. Those who consumed alcohol had a lower intake of calcium and three saturated fatty acids while having a higher intake of two monounsaturated fatty acids. While infants in the study were under the 40th percentile on length, weight, and head circumference at 6 months regardless of alcohol consumption, there were at least 20 nutrients correlated with drinks per drinking day, number of drinking days per week, and/or total drinks per week [30]. As such, nutrition must be considered a preconception- and prenatal-health-related factor that needs to be adequately addressed when supporting pregnant women with alcohol use concerns.

### 3.5. Structural Factors

Structural factors refer to the institutions and the economic, cultural, political, and social structures that perpetuate inequities [55]. Structural factors include policies and laws related to alcohol availability, substance use treatment availability, and the availability of guidance for healthcare professionals [16,18], as well as the systematic attitudes and stigma toward women, pregnant people, mothers, and those who use substances, particularly when pregnant or parenting [18]. In the 2023 literature related to alcohol use in pregnancy, structural factors included a lack of community infrastructure [24,32] and structural support [37,42], stigma [36], inadequate prenatal care [25,41], and inadequate responses in the justice system [33,41]

The lack of community infrastructure and structural support manifested in several ways. For example, in two South African studies, the lack of community infrastructure supportive of alternatives to drinking and the lack of economic opportunities were identified as structural barriers to reducing/stopping alcohol use in pregnancy [24,32]. One study found that in two South African towns with a high FASD prevalence, 57% of respondents expressed concerns about the drinking culture, the lack of hobbies and recreational opportunities, and unemployment-related hopelessness [32]. Other studies explored the lack of structural support for women with lower incomes, migration backgrounds, and who faced other sociocultural challenges [37].

Interpersonal and institutional stigma was also cited as a barrier to reducing/stopping alcohol use in pregnancy and accessing care. For example, another South African study found that pregnant women who drank alcohol in pregnancy reported stigmatizing interactions with healthcare providers and thus discomfort in seeking care at antenatal clinics as barriers to seeking care. In that study, the antenatal service providers interviewed reported that the barriers experienced by pregnant women who drink alcohol manifested through late initiation and/or irregular attendance in antenatal care [36].

These barriers were found to be heightened in institutional settings. For example, a USA study exploring the prevalence and characteristics of incarcerated women with substance use histories found that between 30% and 96% of women who enter prison report using substances during pregnancy, a rate significantly higher than the rates of perinatal substance use found in community samples. The authors reported that enhanced perinatal services are greatly needed to address the needs of mothers with substance concerns in such settings [41]. Reproductive mental healthcare was identified as another system of care that needs to offer and include equitable, treatment-focused, and non-punitive approaches to support pregnant individuals with perinatal substance use concerns. Lastly, the lack of substance use treatment available for women of childbearing age presented an additional barrier to pregnant women/mothers with alcohol use concerns who require treatment. A USA study examined substance use and treatment characteristics of pregnant women and women of child-bearing age (*n* = 97,830) between the ages of 15 and 44. Among women with a substance use disorder, fewer than 13% received treatment regardless of their pregnancy status. The low treatment rates suggest barriers to accessible treatment for all women of child-bearing age [13].

## 4. Discussion

In this article, we categorized the global literature published in 2023 on the influences on maternal alcohol use into five key areas of influence: (1) informational factors; (2) stress-related factors; (3) SDoH-related influences; (4) preconception- and prenatal-health-related factors; and (5) structural factors. In so doing, this article brings clarity to the nature of these influences and the importance of acting on multiple levels.

When FASD prevention was initially described in the 1990s, it was naively suggested that providing information about the risks of prenatal exposure would be sufficient to prevent alcohol use in pregnancy, because there was the assumption that having information about the risks would prompt motivation for alcohol abstinence in pregnancy [56]. Over the decades, we have seen how the provision of information on risks is very necessary, but insufficient as a standalone approach [12]. The research described in this article shows how important, yet inadequately addressed, the informational factors influencing alcohol use in pregnancy remain. Further, it reinforces that it is not informational factors alone, but informational, stress-related, preconception- and prenatal-health-related, and structural factors that need to be addressed to reduce prenatal alcohol use.

The summarized literature indicates that we need to address a lack of information and knowledge about the risks of perinatal alcohol use for both maternal and fetal health [27,31,44], inaccurate beliefs about the risks, stereotypes about who is at risk [27], and a lack of knowledge about where one can safety disclose and seek support [33,36]. The work to provide accurate information about the risks of perinatal alcohol use, in a way that can be heard and acted on at the individual, community, and societal levels, continues to be needed. These efforts must be acted on in a way that links informational needs with other key areas of influence.

Persistent stereotypes, as well as the lived realities of those who will continue to use alcohol in pregnancy, also need to be considered and addressed. Thus, in addition to universal public health messaging, tailored communication is warranted to reach diverse women facing different socioeconomic realities [53]. While research and practice often focus on interventions for people with a lower socioeconomic status, this findings in this article demonstrated that both lower and higher socioeconomic status can increase risk for alcohol use in pregnancy. As such, there needs to also be tailored communication to women using alcohol in pregnancy who have higher income levels, in addition women who face socioeconomic challenges, marginalization, and/or may be at risk of receiving stigmatizing and below-standard antenatal care. A range of socioeconomic factors need to be considered in developing tailored responses. While the 2023 literature saw education and maternal age as protective factors, the broader literature has found that education [57,58,59,60,61,62,63] and maternal age [53,57,59,64] are both protective and risk factors for alcohol use in pregnancy [12]. Other SDoH-related factors that need attending to include food [65] and housing security [66]. Pairing action on informational and SDoH-related influences will challenge misperceptions about who is at risk and what forms of intervention will best support their reasons for change [17].

The stress-related influences on alcohol use in pregnancy are often entangled with SDoH-related and structural factors. In 2023, stress-related factors included mental health considerations [23,26,35,43], experiences of interpersonal and/or intimate partner violence, [35,43], social pressure to use alcohol [24], parenting challenges [31], interpersonal conflict [31], and childhood trauma [35]. These findings align with the broader literature, which has demonstrated the profound role of adverse child [67,68], trauma [69,70], residential school histories [71], mental health status [65,72,73], familial conflict [72], external stressors (e.g., societal pressure, financial strain) [74,75], and social isolation [60,76] on alcohol use in pregnancy. Research has also increasingly attended to the use of alcohol as a coping mechanism to help deal with life stressors [77] or violence and abuse [72,78,79].

When alcohol and other substances are used to cope with depression, anxiety [23,26,35], and the lasting effects of experiences of trauma [69,70] and violence [72,78,79], FASD prevention efforts need to include health-promotion messages and support that are inclusive of the many systems where women are likely to access care beyond perinatal care, including housing, mental health, anti-violence, cultural, recreational, and vocational services. Trauma-informed approaches have been identified as one such promising practice for addressing the stress-related influences on women’s alcohol use and reducing stigma towards women who use alcohol during pregnancy [50]. By doing so, the root influences on alcohol use, the reasons for change can be discussed, and finding alternative ways to cope can be explored [17]. Evidence supports the need for increased commitment by healthcare and social service providers to discuss and address alcohol and other substance use in a multitude of welcoming settings that serve women [12]. This is being achieved in models where cross-sectoral collaborations and partnerships are used to offer wraparound support [80,81].

In 2023, preconception and perinatal health factors such as higher parity [25,35], preconception substance use [39,45], and smoking during pregnancy [25] were most frequently cited as being associated with alcohol use in pregnancy. In our previous research collating the state of FASD prevention research from 2015–2021 [12], we found that partner use and pre-pregnancy alcohol consumption remained two key risk factors for prenatal alcohol consumption. As such, it is integral that FASD prevention efforts start before conception and/or pregnancy recognition. Such efforts should be offered by an array of health and social care providers and be connected to multiple issues. For example, brief interventions in the preconception period can focus both on alcohol use and anxiety to help women overcome barriers for change [82]. Preconception interventions, like those offered in the perinatal period, should be accessible, welcoming, trauma-informed, and non-judgmental.

In 2023, a range of factors that influence alcohol use in pregnancy were described. Structural factors such as the lack of visible, accessible treatment and support for those with alcohol problems were least cited in 2023 but need to be acted on, so that the communication of risks can be paired with communication as to where non-judgmental and accessible support and/or treatment can be accessed, for those with all levels of alcohol use severity. These factors were described both among the general population, as well as among subpopulations of pregnant women, such as pregnant women in carceral and treatment settings, those living with HIV, and pregnant Indigenous and Latina women. Within these subpopulations, stigma being cited as a barrier to service access is particularly relevant for diverse women. Previous research has found that types of stigma, including self-stigma, guilt, and low self-efficacy [74], interpersonal stigma, institutional stigma, and population-level stigma that constructs perceptions around ‘good’ and ‘responsible’ mothering, remain pervasive influences on perinatal alcohol use [18]. Further, research has demonstrated the need for better collaboration across the health and child welfare fields, including the risk of relapse following child removal at birth—a factor that is more commonly articulated in the research on perinatal and maternal substance use [83]. Cross-system collaboration, systems of care, and policies can help redress the barriers that prevent women from seeking care, because they can better respond to the range of factors that contribute to perinatal alcohol use [16]. Systems of care and policies that offer trauma-informed, culturally safe, and strength-based support [17] and that are supportive of mothering [16] are vitally needed.

### Limitations

There has been extensive research published on the prevalence and factors associated with alcohol use in pregnancy. Between 2015 and 2024, over 300 peer-reviewed English-language articles have been published on the prevalence of and influences and factors related to alcohol use in pregnancy. This article only captures a subset of the literature (*n* = 27 peer-reviewed, English-language articles); however, the factors articulated in this narrative review are representative of what has been found in systematic reviews and meta analyses published both prior to and since 2023 [18,47,48,84,85,86].

An additional limitation of this work is the use of a narrative review method. Narrative reviews encourage researchers to describe what is known about a topic while simultaneously offering examination and critique [20]. In doing this, this article offered five key categories of influence that fellow researchers, policymakers, and program planners can consider when developing women’s health and FASD prevention programs. The five key categories of influence identified in the thematic analysis emerged from discussions of this body of literature by the authors and members of the Prevention Network Action Team. While there may be fluidity in these categories, the five identified categories of influence align with the broader literature on the influences and factors associated with alcohol use in pregnancy, including the findings from a Delphi process to construct the Pregnancy Alcohol Use Risk Perception (PARP), which categorized the influencing factors as individual, sociocultural, and informational, and latter included consumption-related factors, determinants, sociocultural norms, and institutional/political/organization influencing factors [87]. A final limitation is that the search used in the narrative review was limited to six databases using EBSCOhost, which may have limited the quantity and quality of the identified articles for inclusion. Future research is needed that systematically explores how these categories of influence can be used to guide health-promotion and FASD prevention programming.

## 5. Conclusions

In this article, we summarized the global literature published in 2023 on the factors that influence alcohol use in pregnancy and categorized them into five key areas of influence. Through understanding the influences as informational, stress-related, SDoH-related, preconception- and prenatal-health-related, and structural, our understanding of perinatal alcohol use can go beyond individual, behavioralist approaches and allow us to develop comprehensive, yet more focused and effective, approaches to FASD prevention. Organizing the influences into categories allows those undertaking prevention efforts to see the influences clearly and to consider how they interact. In doing so, we demonstrate that it is not solely health information, but rather a range of health promotion initiatives, public health programs, substance use and mental health treatments, and policy efforts that is needed to reduce alcohol use in pregnancy. Moving forward, it is integral to address these influences—as broad and complex as they are—in concerted and integrated rather than siloed ways.

## Figures and Tables

**Table 1 ijerph-22-01309-t001:** Overview of included studies (*n* = 27).

Author	Title	Method	Country	Informational Factors	Stress-Related Factors	SDoH-Related Influences	Preconception/Prenatal Health-Related Factors	Structural Factors
Bete et al. [23]	Alcohol consumption and associated factors among pregnant women attending antenatal care at governmental hospitals in Harari regional state, Eastern, Ethiopia	Cross-Sectional	Ethiopia		x	x	x	
Brittain et al.[24]	Perinatal alcohol use among young women living with HIV in South Africa: Context, experiences, and implications for interventions	Qualitative	South Africa	x	x	x	x	x
Broccia et al. [1]	Heavy prenatal alcohol exposure and obstetric and birth outcomes: a Danish nationwide cohort study from 1996 to 2018	Cross-Sectional	Denmark		x	x	x	
Cabral et al.[25]	Prevalence of alcohol use during pregnancy, Brazil, 2011–2012	Cross-Sectional	Brazil			x	x	
David et al.[26]	Exploring the associations between serious psychological distress and the quantity or frequency of tobacco, alcohol, and cannabis use among pregnant women in the United States	Cross-Sectional	US		x		x	
Fletcher et al. [27]	Is ‘a little’ too much?: An exploration of women’s beliefs about alcohol use during pregnancy	Cross-Sectional	Australia	x				x
Fletcher et al. [28]	Intention to engage in alcohol use during pregnancy: The role of attitudes and prototypes	Randomized Controlled Trial	Australia and UK	x				
Green et al. [13]	Substance use and treatment characteristics among pregnant and non-pregnant females, 2015–2019	Cross-Sectional	Canada				x	x
Hamutenya and Nghitanwa [29]	Practices of pregnant women regarding tobacco and alcohol use during pregnancy at one primary health care clinic in Southern Namibia	Mixed-Methods	Namibia			x	x	
Hasken et al. [30]	Maternal dietary intake among alcohol-exposed pregnancies is linked to early infant physical outcomes in South Africa	Cross-Sectional	South Africa				x	
Hernandez et al. [31]	Reasons and obstacles for changing risky drinking behavior among Latinas at risk of an alcohol-exposed pregnancy	Qualitative	US	x	x	x	x	
Jordan et al. [32]	Rethinking local economic development for Fetal Alcohol Spectrum Disorder in Renosterberg Local Municipality, South Africa	Mixed-Methods	South Africa		x	x		x
Kim et al. [33]	Substance use among pregnant women in NSW prisons	Cross-Sectional	Australia	x		x		
Lange et al. [34]	Antenatal care and health behavior of Pregnant women: An evaluation of the survey of neonates in Pomerania	Cross-Sectional	Germany	x		x	x	
May et al. [35]	Maternal risk factors for fetal alcohol spectrum disorders: Distal variables	Case–Control	South Africa	x	x	x	x	
Petersen Williams et al. [36]	Community-based counselling programme for pregnant women with alcohol problems in Cape Town, South Africa: A qualitative study of the views of pregnant women and healthcare professionals	Qualitative	South Africa	x	x	x	x	
Römer et al. [37]	Alcohol and nicotine consumption during pregnancy: Prevalence and predictors among women in Bremen, Germany	Cross-Sectional	Germany			x	x	x
Ruyak et al. [38]	Impulsivity and alcohol use during pregnancy and postpartum: Insights from novel methodological approaches within the context of the COVID-19 pandemic	Cross-Sectional	US		x			
Schmidt et al. [39]	A harmonized analysis of five Canadian pregnancy cohort studies: Exploring the characteristics and pregnancy outcomes associated with prenatal alcohol exposure	Cross-Sectional	Canada				x	
Stanton et al. [40]	Factors associated with changes in alcohol use during pregnancy and the postpartum transition among people with HIV in South Africa and Uganda	Cross-Sectional	South Africa & Uganda			x		
Steely Smith et al. [41]	An integrative literature review of substance use treatment service need and provision to pregnant and postpartum populations in carceral settings	Review	US			x		x
Stevenson et al. [42]	The global burden of perinatal common mental health disorders and substance use among migrant women: A systematic review and meta-analysis	Systematic Review	UK		x		x	
Sujan et al. [43]	Patterns of substance use during early pregnancy and associations with behavioral health characteristics	Cross-Sectional	US		x		x	
Taylor et al.[44]	Accounts of women identified as drinking at ‘high risk’ during pregnancy: A meta-ethnography of missing voices	Meta-Ethnography	UK	x	x			
Thomson et al. [45]	Continuities in maternal substance use from early adolescence to parenthood: Findings from the intergenerational cohort consortium	Cohort	Australia				x	
Tigka et al.[46]	Maternal tobacco, alcohol and caffeine consumption during the perinatal period: A prospective cohort study in Greece during the COVID-19 pandemic	Cross-Sectional	Greece					
Wogayehu et al. [47]	The epidemiology of khat (catha edulis) chewing and alcohol consumption among pregnant women in Ethiopia: A systematic review and meta-analysis	Systematic Review	Ethiopia		x	x	x

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
