# Peer review of "Health Promotion and Support Grounded in Interconnected Influences on Alcohol Use in Pregnancy"

_ijerph, 2025, doi:10.3390/ijerph22081309_

Round 1
Reviewer 1 Report
Comments and Suggestions for Authors
FASD is a global public and devastating human health problem. This systematic review highlights categories of potential preventive and therapeutic interventions and shows the broad complexity of factors contributing to harmful behaviors in pregnant women. Several areas of clarification could help improve the impact of this work, which, for the most part, reveals no surprises.
- It would be of interest to know how non-drinking women manage stress during pregnancy.
- There is no information about macronutrient intake in mothers with FASD offspring. Dietary adjustments can be achieved. Should there be a "Fertile Women's diet” to capture unexpected pregnancies?
- Although mention is made of middle-class or wealthy women's drinking--their habits in comparison with lower SES mothers could be better included.
- Public health measures to halt drinking in pregnancy abound. What are the trends, year-over-year, since the messaging was initiated?
- It would be great if the authors could prioritize how to address drinking by ranking the correlational factors. The societal problems are utterly complex and have no easy fix. However, are there measures that could be taken to help protect fetuses and infants in resource-poor environments?
- The entire message is complex and difficult to tackle in any practical manner. Pre-pregnancy behaviors in women are difficult to address. The stress links to alcohol consumption should be made evident and distinguished from regular stresses in the environment as endured by non-alcohol-consuming pregnant women. Behavioral responses, e.g., binge drinking, smoking, and low education, are evident, but are the initial and ongoing stresses pertaining to social pressures, childhood trauma, partner violence, etc., greater in women who consume excessive alcohol during pregnancy?
Reviewer 2 Report
Comments and Suggestions for Authors
Authors have picked an interesting topic; however, the content and method are not as significant as the title. We suggest improving the methodology using a systematic literature review rather than a narrative review to strengthen the message.

Author Response
Although this study has a good aim, but it need a strong and update evidence to support the results. |
The methods have been updated to clarify the five categories of influence used to analyze the factors that influence prenatal alcohol use. |
Is this a narrative review or systematic review? |
This is a narrative review. |
Why not to 2025? why it has to be in the 2023 year? |
From 2015 – 2024, there were over 300 articles English-language articles published on the prevalence of, and influences and factors related to alcohol use in pregnancy; the majority of which focused solely on the influences and factors. By narrowing the scope to articles from 2023 only, we were able to concisely capture a snapshot of the range of factors that influence alcohol use in pregnancy. You’ll see from the discussion, that these five categories apply also to the greater literature on the factors and influences related to prenatal alcohol use. |
Keywords: double check with the MeSH guidelines |
The MeSH terms have been updated. |
Update final paragraph of the introduction so that the paragraph consists of three sentences. |
This has been updated. |
Describe in the figure for the flowchart of the article selection |
A flowchart is not typically used in a narrative review, and thus we have not included it in the revisions. |
In the methods, clarify who was discussed for consensus |
This has been updated. |
Add N study information in the bracket (N=xx) in the Table |
This has been updated. |
Update paragraph with structural factors (discussion) to include three sentences |
This has been updated. |
Update the conclusion to be more concise |
This has been updated. |
Reviewer 3 Report
Comments and Suggestions for Authors
The author limited the literature search and review to the period from January to December 2023. What is the reason for this approach? Why were relevant studies before 2023 not included—is it because there were already existing literature reviews on them? If so, there should also be numerous relevant studies from 2024 to 2025. Why weren't they included either?
The authors searched six databases using EBSCO Host, including Bibliography of Native North Americans, CINAHL (Cumulative Index of Nursing and Allied Health Literature) Complete, MEDLINE with Full Text, PsycINFO, Social Work Abstracts, and Urban Studies Abstracts. However, key databases such as Embase, Web of Science, Scopus, and Cochrane Library were not included, which may significantly limit the quantity and quality of the retrieved literature, potentially affecting the study's conclusions.
Additionally, the authors did not clearly describe their search strategy and procedures. Critical details—such as the specific keywords used, the exact search methodology, the initial number of identified records, the number of excluded studies, and the final count of articles included in the analysis—were not sufficiently reported. To ensure a clear presentation of the research process and methodology, the authors should include a flowchart like PRISMA to visually illustrate the key steps and procedures.
The authors identified five distinct categories of influencing factors. What were the specific criteria and rationale behind this classification? Why were these factors grouped into five types rather than four or six?
Additionally, is this review a systematic review? If so, systematic reviews require adherence to a standardized evaluation process, including the PICOS/PECO framework, rigorous literature screening, risk of bias assessment, structured reporting of results, and a clear discussion of limitations.
Furthermore, the authors failed to acknowledge the limitations of this study or propose potential measures for improvement.
The authors summarized five distinct influencing factors through their literature review and identified specific elements within each category, which undoubtedly holds some significance. However, the issue lies in the fact that the conclusions of a review study are inherently determined by the quantity and quality of the literature searched. In this case, the study was limited to literature published only within the narrow timeframe of 2023. As a result, the conclusions drawn may only be applicable to this exceptionally brief period, significantly diminishing the broader relevance of the research. The authors should clearly justify why they restricted their review to just one year or, alternatively, substantially expand both the temporal scope and range of the literature search.
Round 2
Reviewer 1 Report
Comments and Suggestions for Authors
The authors did an excellent job responding to the critiques.
Author Response
no further comments required
Reviewer 2 Report
Comments and Suggestions for Authors
We accepted this current version.
Author Response
No further comments required
Reviewer 3 Report
Comments and Suggestions for Authors
The revised version does not directly address the reviewers' core concerns. e.g.
"The author limited the literature search and review to the period from January to December 2023.What is the reason for this approach? Why were relevant studies before 2023 not included—is it because there were already existing literature
reviews on them? Simutlaneously, there should also be numerous relevant studies from 2024 to 2025. Why weren't they included either?"
"The authors searched six databases using EBSCO Host, including Bibliography of Native North Americans, CINAHL (Cumulative Index of Nursing and Allied Health Literature) Complete, MEDLINE with Full Text, PsycINFO, Social Work Abstracts, and Urban Studies Abstracts. However, key databases such as Embase, Web of Science, Scopus, and Cochrane Library were not included, which may significantly limit the quantity and quality of the retrieved literature, potentially affecting the study's conclusions."
"The authors identified five distinct categories of influencing factors. What were the specific criteria and rationale behind this classification? Why were these factors grouped into five types rather than four or six?"
"The authors summarized five distinct influencing factors through their literature review and identified specific elements within each category, which undoubtedly holds some significance. However, the issue lies in the fact that the conclusions of a review study are inherently determined by the quantity and quality of the literature searched. In this case, the study was limited to literature published only within the narrow timeframe of 2023. As a result, the conclusions drawn may only be applicable to this exceptionally brief period, significantly diminishing the broader relevance of the research. The authors should clearly justify why they restricted their review to just one year or, alternatively, substantially expand both the temporal scope and range of the literature search."
Unless these core issues are resolved, they will severely compromise the study's conclusions and quality. Unfortunately, the revised manuscript completely ignored these issues. As a reviewer, I find that this revision evades critical academic concerns, and the modifications made contribute little to improving the paper's quality.
Author Response
The author limited the literature search and review to the period from January to December 2023.What is the reason for this approach? Why were relevant studies before 2023 not included—is it because there were already existing literature
reviews on them? Simutlaneously, there should also be numerous relevant studies from 2024 to 2025. Why weren't they included either?"
We have tried to address this in the discussion and limitations section - the manuscript emphasizes how, despite using a subset of literature (from the year 2023), the findings of this review and the five key categories of influence are responsive and representative to the broader literature on factors influencing alcohol use in pregnancy
"The authors searched six databases using EBSCO Host, including Bibliography of Native North Americans, CINAHL (Cumulative Index of Nursing and Allied Health Literature) Complete, MEDLINE with Full Text, PsycINFO, Social Work Abstracts, and Urban Studies Abstracts. However, key databases such as Embase, Web of Science, Scopus, and Cochrane Library were not included, which may significantly limit the quantity and quality of the retrieved literature, potentially affecting the study's conclusions." We have used these databases in our annual literature searches over the past decade. They have served well to help us identify sociocultural influences on pregnant women's health. We recognize that more medical model and clinical databases could have been included.
"The authors identified five distinct categories of influencing factors. What were the specific criteria and rationale behind this classification? Why were these factors grouped into five types rather than four or six?" We have tried to address this in the discussion and limitations sections. These categories have gradually become apparent to us, and the studies in the 2023 year served to crystallize them.